# Effects of Wormwood (*Artemisia montana*) Essential Oils on Digestibility, Fermentation Indices, and Microbial Diversity in the Rumen

**DOI:** 10.3390/microorganisms8101605

**Published:** 2020-10-18

**Authors:** Seong Shin Lee, Dong Hyeon Kim, Dimas Hand Vidya Paradhipta, Hyuk Jun Lee, Hee Yoon, Young Ho Joo, Adegbola T. Adesogan, Sam Churl Kim

**Affiliations:** 1Division of Applied Life Science (BK21Four, Institute of Agriculture and Life Science), Gyeongsang National University, Jinju 52828, Korea; seongshin73@gmail.com (S.S.L.); dimazhand@gmail.com (D.H.V.P.); hyukjun0209@gmail.com (H.J.L.); yoonhee8724@gmail.com (H.Y.); wn5886@gmail.com (Y.H.J.); 2Department of Animal Sciences, IFAS, University of Florida, Gainesville, FL 32611, USA; adesogan@ufl.edu; 3Dairy Science Division, National Institute of Animal Science, Cheonan 31000, Korea; kimdh3465@gmail.com; 4Faculty of Animal Science, Universitas Gadjah Mada, Yogyakarta 55218, Indonesia

**Keywords:** essential oil, in vitro rumen fermentation, rumen microbe, wormwood

## Abstract

This study investigated the effects of essential oil (EO) from three Korean wormwood (*Artemisia Montana*) plants on in vitro ruminal digestibility, fermentation, and microbial diversity. Dried (0.5 g) soybean meal (SBM) or bermudagrass hay (BGH) were incubated in buffered rumen fluid (40 mL) for 72 h with or without EO (5 mg/kg) from Ganghwa (GA), Injin (IN), or San (SA) wormwood (Experiment 1). Both SA and IN improved (*p* < 0.05) dry matter digestibility (DMD) of BGH, while GA reduced (*p* < 0.05) total short-chain fatty acid of BGH and SBM. Besides, SA increased (*p* < 0.05) numbers of *Ruminococcus albus* and *Streptococcus bovis* in SBM. Experiment 2 examined different doses (0, 0.1, 1, and 10 mg/kg) of SA, the most promising EO from Experiment 1. Applying SA at 10 mg/kg gave the highest DMD (L; *p* < 0.01) and neutral detergent fiber (Q; *p* < 0.05) digestibility for BGH. Applying SA at 1 mg/kg gave the highest *R. albus* population (Q; *p* < 0.05) in SBM. Therefore, SA was better than GA and IN at improving rumen fermentation, and the 0.1 to 1 and 10 mg/kg doses improved ruminal fermentation and in vitro digestibility of SBM and BGH, respectively.

## 1. Introduction

Interest in alternative feed additives has increased over the last few decades since many countries, including the European Union and the United States of America, imposed mandatory or voluntary restrictions on the use of antibiotics for animal production due to adverse effects on human health. Essential oils (EO), which have antimicrobial and antioxidant activity, have been studied extensively as feed additives to replace antibiotics in livestock diets. In general, EO can modify ruminal microbial fermentation and improve ruminant performance [1]. The EO is composed of secondary plant metabolites with different active compounds and concentrations depending on the source [2]. Application of EO as feed additives has varied effects on rumen fermentation. For example, the application of EO from lemongrass oil has improved nutrient digestibility in the rumen [3]. Other sources of EO, such as oregano, thyme, and sweet orange, have been reported to have no effect on nutrient digestibility, ammonia-N, and total short-chain fatty acid (SCFA) of the rumen according to Benchaar et al. [4], but they had beneficial effects on rumen fermentation according to Calsamiglia et al. [1]. In addition, the application of herbal mixtures as EO sources has improved nutrient digestibility and total SCFA concentration but is less effective at reducing methane emission [5]. Furthermore, the application of a mixture of medical plants (wormwood (*Artemisia absinthium* L.), chamomile (*Matricaria chamomilla* L.), fumitory (*Fumaria officinalis* L.), and mallow (*Malva sylvestris* L.)) as EO sources has decreased ammonia-N and methane concentration in vitro [6].

Plant extracts with bioactive compounds have been used as feed additives and rumen modifiers to increase the performance of ruminants [5,6]. Specific Korean wormwood (*Artemisia montana*) has been recognized in many parts of the world as a medicinal and functional food, and its EO has been extracted and characterized. Kim et al. [7] reported that EO from Korean wormwood hay contained Camphor, 1-Borneol, and Caryophyllene Oxide as the main active compounds, whereas Korean wormwood silage contained 3-Cyclohexen-1-o1, t*rans*-Caryophyllene, and ɣ-Selinene. In an animal trial, replacing dietary rice straw with 50–150 g kg^−1^ dry matter (DM) of wormwood hay or silage has increased feed intake, in vivo digestibility, N retention, and microbial N yield in sheep [8,9]. Generally, there are three species of native wormwood that grow widely in South Korea, including Ganghwa, Injin, and San. However, the effects and optimal application level of EO from these native wormwoods have not been studied. According to previous studies [1,2], the optimum level of EO application for ruminants could vary from 0.1 to 10 mg/kg, depending on their active compounds. Studies on the efficacy and optimal dose of EO from the native wormwoods could lead to the development of a native feed additive for improving ruminant performance in South Korea. Therefore, two in vitro study experiments were conducted. Experiment 1 aimed to extract and identify EO from three native Korean wormwood species and to examine their effects on in vitro ruminal fermentation, digestibility, and fibrolytic bacterial counts following bermudagrass or soybean meal fermentation. Experiment 2 aimed to investigate the optimal inclusion levels of the most promising wormwood EO from Experiment 1 for improving in vitro rumen digestibility, fermentation, and fibrolytic bacterial counts on bermudagrass or soybean meal substrates. We hypothesized that the wormwood species and application levels would have varying effects on rumen microbial fermentation.

## 2. Materials and Methods

### 2.1. Substrate and Essential Oil

The low and high fiber substrates consisted of soybean meal (SBM) and bermudagrass hay (BGH, Cynodon dactylon cultivar Tifton 85), respectively. The concentrations of DM and neutral detergent fiber (NDF) of SBM were 889 g kg^−1^ and 93.0 g kg^−1^, respectively, and those for BGH were 925 g kg^−1^ and 664 g kg^−1^, respectively. Those substrates were obtained from the Dairy Research Unit, the University of Florida, dried at 60 °C for 48 h, and ground through a 1-mm screen using a Willey mill (Arthur H. Thomas, Philadelphia, PA, USA) for later use as in vitro fermentation substrates. The EO was extracted from three different Korean native species of wormwoods—Ganghwa (GA), Injin (IN), and San (SA). The raw SA was collected from Jinju, Gyeongnam province, while the raw GA was collected in Ganghwa, Incheon province, South Korea. The powder IN was obtained from Puleun-won Company (Yeongcheon, Gyeonggi province, Korea). Each wormwood was extracted by steam distillation, as described by Maarse and Kepner [10], using the distillation and heating apparatus (Heating Mantle, M tops, Seoul, Korea) described by Likens and Nickerson [11]. The identification of EO composition for each wormwood spp. was determined following the protocol of Kim et al. [7] with a gas chromatograph (Hewlett Packard GC-5890A, Hewlett-Packard Company, Wilmington, NC, USA) equipped with a flame ionization detector and a DB-5 column (60 mm × 0.32 mm × 0.25 µm) coupled to a mass spectrometer (Joel JMS-700, Joel Ltd., West Yorkshire, UK).

### 2.2. In Vitro Incubation

The animal care procedure was approved by the Institutional Animal Care and Use Committee of the University of Florida, USA. Two rumen cannulated non-lactating Holstein cows (40 mo of age) fed 9 kg of coastal bermudagrass (*Cynodon dactylon*) hay, and 400 g of soybean meal daily were used as the inoculum donor. Rumen fluid was collected before morning feeding. Rumen fluid (pH 6.65) was filtered through two layers of cheesecloth and mixed with an anaerobic culture medium at a 1:2 ratio to make buffered rumen inoculum, as described by Adesogan et al. [12]. Dried samples of SBM and BGH, respectively, at 0.5 g were added to 40 mL of buffered rumen inoculum within incubation bottles. In Experiment 1, buffered rumen fluid was treated with or without EO from different native wormwood species to give the following treatments: no EO (Control; CON), GA, IN, and SA. All EO treatments were applied at 5 mg/kg per 40 mL of buffered rumen inoculum. The most effective EO among GA, IN, and SA for improving rumen fermentation indices and population of fibrolytic bacteria was used in Experiment 2.

In Experiment 2, the selected EO from Experiment 1 was applied at the following levels of inoculation: 0, 0.1, 1, and 10 mg/kg per 40 mL of buffered rumen inoculum.

Carbon dioxide gas was flushed into each incubation bottle continuously to maintain anaerobic conditions. Each treatment was prepared in triplicate along with three blanks and incubated at 39 °C for 72 h [13].

### 2.3. Laboratory Analysis

After incubation, buffered rumen fluid (1 mL) from all bottles, including blanks, was sub-sampled for the analysis of microbial diversity using PCR. The remaining buffered rumen fluid after the incubation was filtered through previously dried (60 °C for 48 h) filter paper (Whatman 451 Grade 1, Pittsburgh, PA, USA). The residues on the filter paper were dried for 48 h at 60 °C and weighed. The substrates and dried residues were analyzed for NDF [14] using an Ankom^200^ Fiber Analyzer. Residue weights and NDF concentration were used to calculate in vitro DM digestibility (IVDMD) and in vitro NDF digestibility (IVNDFD), according to a previous study [15]. The supernatant filtered from the in vitro fermentation was used for pH, ammonia-N, and SCFA analyses. The pH and ammonia-N were measured with an electric pH meter (SevenEasy™ pH Meter S20, Mettler Toledo, Greifensee, Switzerland) and a colorimetric procedure described by Chaney and Marbach [16], respectively. The concentrations of SCFA were measured by the method of Muck and Dickerson [17] using an HPLC (L-2130, Hitachi, Tokyo, Japan) coupled with an autosampler (L-2200, Hitachi, Tokyo, Japan), UV detector (L-2400, Hitachi, Tokyo, Japan), and a column (MetaCarb 87H, Varian, Middelburg, The Netherlands).

The sub-sampled buffered rumen fluid supernatant samples reserved for the analysis of microbial diversity were transferred into screw-capped tubes containing silica beads. The samples were shaken using a bead beater (Mini bead beater, Cole-Parmer, Vernon Hills, IL, USA). The DNA was extracted using the QIAamp DNA mini kit (Qiagen, Valencia, CA, USA), according to the manufacturer’s protocol. The DNA concentrations were measured with a NanoDrop Spectrophotometer (ND-1000, Palto Alto, CA, USA). The PCR primer sets used for the amplification of general bacteria, Methanogenic archaea, *Fibrobacter succinogenes*, *Ruminococcus albus*, *Streptococcus bovis*, and *Ruminococcus flavefaciens* are shown in Table 1. Real-time PCR assays were performed on a Bio-Rad C1000 Touch Thermal cycler Real-Time PCR detection system (CFX96 Real-Time System, Bio-Rad Laboratories, Inc., Hercules, CA, USA), according to the methods described by Denman and McSweeney [18]. Assays were set up using an SYBR Green Real-Time PCR Master Mix (TOYOBO Co., Ltd., Osaka, Japan). After real-time PCR, the cycle threshold (Ct) values were used to determine the fold change of different microbial populations relative to the respective blanks without EO application both in Experiment 1 and Experiment 2. The quantity of the microbes was estimated by the equation: relative quantification = 2^-ΔCt (target)- ΔCt (control)^, where Ct represents the threshold cycle. A negative control without the template DNA was used in every real-time PCR assay for each primer. The PCR condition for the amplification of target DNA is shown in Table 1.

### 2.4. Statistical Analysis

All data from Experiment 1 were analyzed with the General Linear Model (PROG GLM) procedure of the Statistical Analysis System (SAS; Version 9. Cary, NC, USA) [22]. The model was Y*_ij_* = µ + T*_i_* + e*_ij_*, where Y*_ij_* = response variable, µ = overall mean, T*_i_* = the effect of EO *i*, and e*_ij_* = error term. All data in Experiment 2 were analyzed with polynomial contrasts using PROG GLM of SAS to examine linear, quadratic, or cubic effects of increasing EO level. Orthogonal coefficients for linear, quadratic, and cubic contrast were adjusted to account for the unequal spacing of EO levels (0 vs. 0.1 vs. 1.0 vs. 10) using the Interactive Matrix Programming Language (PROC IML) procedure of SAS [22]. Mean separation was performed with Tukey’s test, and the significant differences were declared at *p* < 0.05 and tendencies at 0.05 < *p* < 0.10.

## 3. Results

### 3.1. Experiment 1

The main compounds in GA were endo-Borneol (185 g kg^−1^), 3-Cyclohexen-1-o1 (102 g kg^−1^), and Pentan-1,3-Dioldiisobutyrate (511 g kg^−1^), while IN contained Camphor (332 g kg^−1^) and 1-Borneol (299 g kg^−1^) (Table 2). The SA mainly contained *trans*-Caryophyllene (162 g kg^−1^), δ-Cadinene (109 g kg^−1^), α-Cadinol (131 g kg^−1^), and T-Muurolol (151 g kg^−1^).

For the SBM substrate, the application of GA resulted in lower (*p* = 0.026; 14.26 vs. 154.2 m*M* L^−1^) total SCFA concentration than CON, while applying IN and SA had no effect on this measure (Table 3). The IVNDFD of the SBM substrate was completely degraded (100%) in all treatments because it had a low NDF concentration. The application of GA tended to produce higher concentrations of propionate (*p* = 0.097) and valerate (*p* = 0.096) relative to other treatments, while SA tended to produce a higher concentration of isovalerate (*p* = 0.072). For the BGH substrate, the application of IN and SA increased (*p* = 0.004; 440 and 411 vs. 308 and 323 g kg^−1^) IVDMD relative to CON and GA. The application of GA decreased (*p* = 0.001; 217 vs. 370, 356, and 312 g kg^−1^) IVNDFD relative to all treatments. The application of SA resulted in the highest (*p* = 0.001; 20.5 vs. 17.4, 17.0, and 17.9 mgN dL^−1^) ammonia-N concentration, while the application of GA was associated with the lowest (*p* = 0.001; 119.8 vs. 136.1, 130.2, 131.7 m*M* L^−1^) total SCFA concentration.

For the SBM substrate, the *F. succinogens* population was lowest (*p* < 0.001; 2.36 and 2.63 vs. 9.13 vs. 18.8) for IN and SA, followed by CON, and highest for GA (Figure 1). Furthermore, the *R. albus* population was higher (*p* = 0.046; 24.5 vs. 10.8) for SA than CON, while those for GA and IN did not differ compared to other treatments. The *S. bovis* population was highest (*p* < 0.001; 0.61 vs. 0.44 vs. 0.01 and 0.05) for SA, followed by GA, and lowest for CON and SA. The *R. flavefaciens* population was higher (*p* = 0.014; 0.08 and 0.08 vs. 0.04) for GA and IN than CON, while that for SA did not differ compared to other treatments. The methanogenic archaea population was not affected by the application of EO. For the BGH substrate, the population of *R. flavefaciens* was higher (*p* = 0.005; 1.79 vs. 0.53, 0.67, and 0.60) in CON than all EO treatments.

### 3.2. Experiment 2

The SA wormwood was selected for the examination of its optimal inclusion level in Experiment 2 because it was more effective at improving rumen fermentation and increasing fibrolytic bacteria populations than GA and IN when applied at 5 mg/kg per 40 mL of buffered rumen fluid in Experiment 1. As the SA inclusion level increased, the IVDMD (0.05 < *p* < 0.10) and ammonia-N concentration (*p* < 0.01) of SBM followed quadratic and cubic patterns, respectively, with the 0.1 mg/kg level giving the highest value (Table 4). The concentrations of total SCFA (*p* < 0.01), isovalerate (*p* < 0.01), and acetate to propionate (A:P) ratio (*p* < 0.05) increased linearly with increasing SA inclusion, but propionate concentration (*p* < 0.05) decreased linearly.

Increasing SA inclusion linearly increased IVDMD (*p* < 0.01) and butyrate concentration (*p* < 0.05) of BGH, whereas the IVNDFD (*p* < 0.05) exhibited a quadratic pattern with the highest value occurring for the 10 mg/kg level. Acetate concentration (*p* < 0.01) also had quadratic pattern, with the lowest value at the 1 mg/kg level. Concentrations of total SCFA (*p* < 0.01), isobutyrate (*p* < 0.01), and isovalerate (*p* < 0.01) increased in a quadratic manner with the highest values at the 1 mg/kg level. In addition, A:P ratio showed a quadratic pattern with increasing SA level, with the lowest value occurring with the 1 mg/kg dose.

For the SBM substrate, increasing SA inclusion presented a quadratic pattern for *F. succinogens* (0.05 < *p* < 0.1) and *R. albus* (*p* < 0.05) populations (Figure 2). The population of *R. flavefaciens* (*p* < 0.05) also was increased in a quadratic manner by increasing SA levels.

## 4. Discussion

### 4.1. Experiment 1

Application of IN and SA had no effects on rumen fermentation and nutrient digestibility of SBM substrate but increased IVDMD of BGH substrate. Ruminal pH and total SCFA concentration in the present study were similar to the previous studies [13,23,24]. The GA decreased total SCFA concentration resulting from the fermentation of both substrates and decreased also IVDNDF of the BGH substrate. The IVNDFD of the SBM substrate was 100% in the present study because SBM was totally degraded in the ruminal fluid due to its low concentration of NDF. Similar to the increase in IVDMD of BGH by IN and SA, Kim et al. [9] reported that a replacing rice straw with wormwood silage in a diet with a high straw to concentrate ratio (70:30) improved DM digestibility of sheep [9]. However, the species of wormwood used in the study was not specified. In an in vitro study, the application of a mixture of medicinal plants containing common wormwood (*Artemisia absinthium* L.) to a sheep diet has increased total SCFA concentration and reduced ammonia-N concentration [6].

Different active EO compounds have likely contributed to the specific responses on rumen fermentation in the above studies [1,2,3,6]. For example, dietary addition of wormwood as hay or silage has given different rumen fermentation results in sheep, perhaps because the wormwood active compounds differ [8,9]. Kim et al. [7] discovered that wormwood hay contained Carypophyllene Oxide, 1-Borneol, Camphor, and *trans*-Caryophyllene as the major active compounds, while wormwood silage contained γ-Selinene and 3-Cyclohexen-1-ol. In some previous studies, EO has provided an antibacterial activity that could decrease pathogenic bacteria populations and increase feed efficiency [1]. However, some active compounds of EO could also inhibit rumen fermentation [2]. In the present study, different wormwood species had different active compounds (Table 2) that likely contributed to the different results on rumen digestibility, fermentation, and microbial diversity. More research investigating the active compounds of each wormwood species and how they are affected by the preservation method is warranted to develop feed additives from wormwood EO.

As for IN and SA in the present study, previous studies have reported the potential for EO from lemongrass oil, cinnamon oil, or clove oil to improve nutrient digestibility and rumen fermentation [1,3]. Petrič et al. [6] also applied a mixture of medical plants, including wormwood, in an in vitro study and improved ammonia-N and total SCFA concentrations, as found for IN and SA in the present study. However, other studies have also reported a similar result to that from GA, which reduced the digestibility of BGH and fermentation of both substrates [2,25,26]. The GA contained Petan-1,3-diodiisobutyrate as the active compound with the highest concentration, whereas IN and SA did not contain this compound. No study has examined the effects of pure Petan-1,3-diodiisobutyrate on rumen fermentation. However, based on its high concentration in GA and the negative effect GA had on in vitro digestibility, it is assumed that Petan-1,3-diodiisobutyrate has an inhibitory or negative effect on rumen fermentation.

The IN wormwood contained similar major active compounds (Camphor and 1-Borneol) as that of wormwood hay (*Artemisia montana* Pampan; *Artemisia* sp.) [7,8]. Substitution of rice straw with wormwood hay containing Camphor and 1-Borneol in sheep diets has increased total ammonia-N and total SCFA of ruminal fermentation relative to the control diet within 2 h of feeding, but the difference is no longer evident after 8 h of feeding [8]. In the present study, IN had a similar SCFA concentration to CON after 72 h of in vitro incubation.

Effects of the main active EO compounds in SA, including *trans*-Caryophyllene, δ-Cadinene, α-Cadinol, or T-Muurolol, on rumen fermentation are unknown. The α-Cadinol and T-Muurolol in SA are also found in the EO of other medicinal plants, such as *Alphinia zerumbet* (shell ginger) and *Neolitsea fischeri* [27,28], but those studies have focused on antibacterial and antioxidant effects. In general, all of the major active compounds in IN and SA have been reported to inhibit pathogenic bacteria, such as *Bacillus cereus*, *Staphylococcus aureus*, *Escherichia coli*, *Pseudomonas flourescens*, and *Listeria monocytogenes* [28,29].

All EO treatments applied to the SBM substrate decreased the population of *F. succinogens* but increased those of *R. albus*, *S. bovis*, and *R. flavefaciens*. All EO treatments applied to BGH decreased the *R. flavefaciens* population. According to Lin et al. [30], the populations of *F. succinogenes* and *R. flavefaciens* decreased with the application of a combination of different EO (thyme oil, oregano oil, cinnamon oil, and lemon oil) to a mixed substrate containing ground maize and *Leymus chinensis* hay (1:1 ratio). Unfortunately, specific contributions of each EO type were not documented. In another study, the application of single EO, such as clove oil, eucalyptus oil, garlic oil, origanum oil, and peppermint oil, to a mixed substrate containing alfalfa and concentrate (1:1 ratio) has also decreased the populations of *F. succinogenes* and *R. flavefaciens* [31]. These previous studies have indicated that the application of EO has negative effects on *F. succinogens* and *R. flavefaciens* populations, which supports the results of the EO in this study on SBM and BGH substrates, respectively. Nevertheless, certain fibrolytic bacteria populations were increased by certain EO, which indicates that different bacteria had different susceptibility to different EO. For example, in the SBM substrate, the application of SA led to the highest increases in the populations of *R. albus* and *S. bovis*, while GA and IN increased the population of *R. flavefaciens*. The different active compounds of these wormwood species (Table 2) may explain the differential results on specific fibrolytic rumen bacteria. Although the present study showed that the effectiveness of EO to improve rumen fibrolytic bacterial populations depended on the wormwood spp. and substrate, more research is needed to understand which specific wormwood EO components have the beneficial and deleterious effects.

In general, SA was the most effective wormwood spp. tested because it increased IVDMD and IVNDFD of BGH and increased the population of *R. albus* and *S. bovis* in the SBM when it was fermented without decreasing total SCFA concentration. Therefore, it was selected for Experiment 2 to examine the optimum level of application.

### 4.2. Experiment 2

As in Experiment 1, increasing the level of inclusion of SA influenced rumen fermentation [31]. The application of SA at 0.1 mg/kg to the SBM substrate tended to produce a higher IVDMD without any negative effects on the SCFA profile and gave the highest ammonia-N concentration. This study, therefore, supported the use of this level (0.1 mg/kg) to improve IVDMD [32]. The application of SA at 10 mg/kg to SBM increased total SCFA concentration but reduced IVDMD. This reduction in IVDMD might have been due to the inhibition of microbial growth by the high level of EO supplied, even though no effects on the fibrolytic bacteria studies were evident. Other studies have shown that an appropriate dose of EO can increase the fermentation of substrates in the rumen, whereas excessive doses can decrease rumen fermentation [2,33]. For instance, Busquet et al. [23] reported that adding plant extract EO at 3–300 mg L^−1^ to a mixed substrate (1:1 ratio of forage and concentrate) increased total SCFA and ammonia-N concentration, but they were decreased by applying EO at 3000 mg L^−1^. Likewise, Castillejos et al. [24] reported that adding various EO at 5–500 mg L^−1^ to a mixture of forage and concentrate (60:40 ratio) increased concentrations of total SCFA and ammonia, but these were reduced at a high dose of 5000 mg L^−1^. More research is needed to understand specific reasons and mechanisms by which high doses of EO inhibit rumen fermentation.

The application of SA EO at different levels to SBM had no effects on fibrolytic bacteria populations in this study for unknown reasons. The application of 1 mg/kg of SA to SBM substrate increased the populations of *F. succinogenes* and *R. albus*, while the application of 10 mg/kg SA decreased them. This result was similar to previous studies, which have shown that appropriate doses of EO could increase fibrolytic bacteria populations, but excessive doses inhibit their growth [1,2,31]. A previous in vivo study has also reported that increasing the level of a commercial EO mixture containing thymol, eugenol, vanillin, guauacol, and limonene to 150 mg kg^−1^ concentrate in the diet of dairy ewes has increased fibrolytic bacteria population in the rumen [23]. In addition, Calsamiglia et al. [1] and Benchaar et al. [2] reported that the growth of different fibrolytic bacteria varied with the level of EO applied. This may explain why growth responses of *S. bovis* and *R. flavefaciens* differed from those of *F. succinogenes* and *R. albus* when different doses of SA EO were examined in the present study. In addition to the dose, the specific individual and combined effects of active EO compounds likely contribute to modulating the growth of the fibrolytic bacteria [1,2].

## 5. Conclusions

In the present study, the application of SA wormwood had more beneficial effects than GA and IN wormwood as it increased in vitro DMD of BGH and increased the population of *R. albus* and *S. bovis* when the SBM was fermented. The major active compounds in SA were *trans*-Caryophyllene, δ-Cadinene, α-Cadinol, and T-Muurolol. In addition, the applications of SA at 0.1 mg/kg increased the rumen fermentation and in vitro DMD of SBM, while the 1 mg/kg dose increased the population of *F. succinogenes* and *R. albus* when the SBM was fermented. The 10 mg/kg dose of SA increased the in vitro DMD of BGH, while the 1 and 10 mg/kg doses increased in vitro fermentation.

## Figures and Tables

**Figure 1 microorganisms-08-01605-f001:**
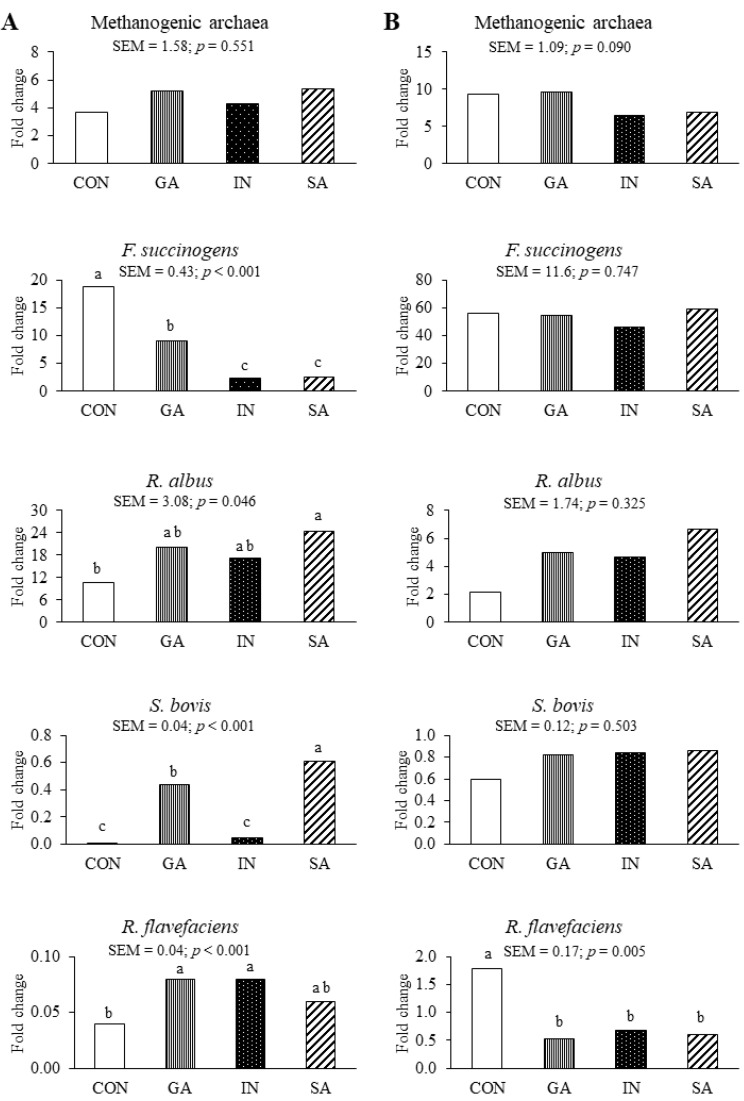
Effects of adding essential oils from different wormwood species to soybean meal (SBM) (**A**) or bermudagrass hay (BGH) (**B**) on fold changes of rumen microbial populations compared to that of the blank after 72 h of in vitro incubation in buffered rumen fluid. CON, without essential oil application; GA, Ganghwa essential oil applied; IN, Injin essential oil applied; SA, San essential oil applied. a–c: Means for the same microbe with different letters differ significantly (*p* < 0.05).

**Figure 2 microorganisms-08-01605-f002:**
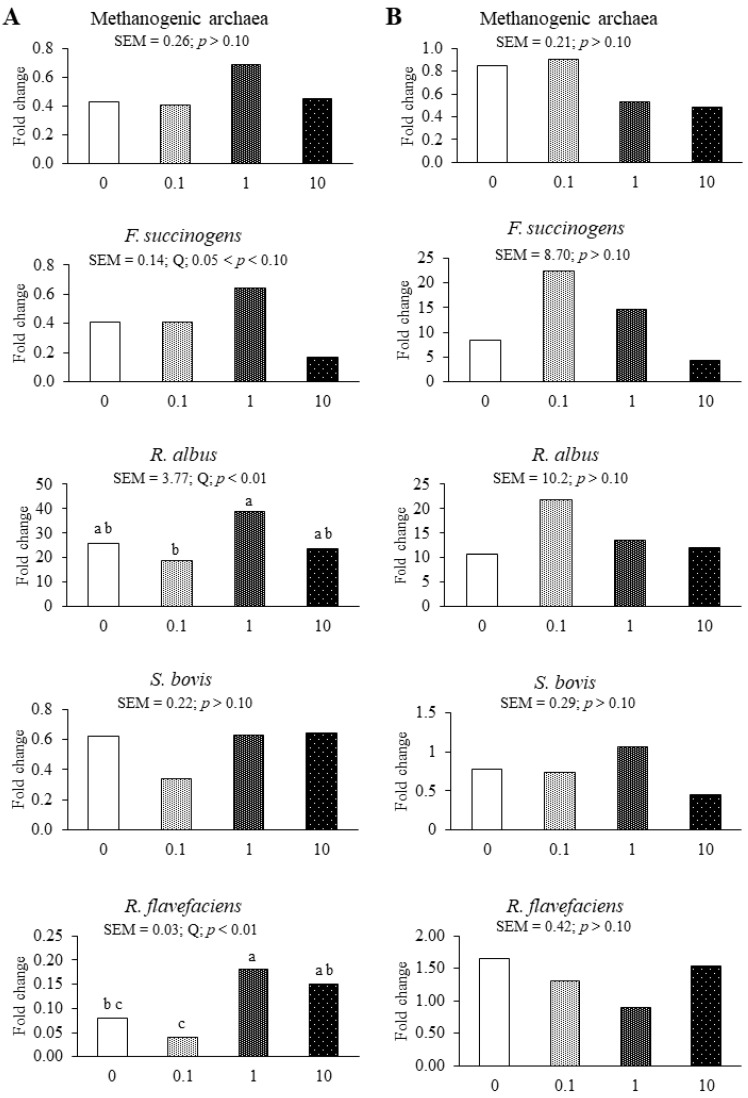
Effects of applying essentials oil from San wormwood at increasing levels to SBM (**A**) or BGH (**B**) substrates on fold changes of rumen microbial populations compared to that of the blank after 72 h of in vitro incubation in buffered rumen fluid. The San essential oil was applied at 0, 0.1, 1, and 1 mg/kg per 40 mL of buffered rumen fluid. The *p*-value indicated the polynomial contrast analysis; L, Linear effect; Q, Quadratic effect; C, Cubic effect. a–c: Means for the same microbe with different letters differ significantly (*p* < 0.05).

**Table 1 microorganisms-08-01605-t001:** List of Primers and PCR Conditions for Rumen Microbial Assay.

Target Species	Primers	Size (bp)	PCR Condition (*modified from the ref.)	Reference
Methanogenic archaea	TTCGGTGGATCDCARAGRGC/ GBARGTCGWAWCCGTAGAATCC	140	*95 °C for 3 min, (95 °C for 15 s, 60 °C for 30 s, 72 °C for 30 s)- 48 cycles.	[19]
*F. succinogens*	GTTCGGAATTACTGGGCGTAAA/ CGCCTGCCCCTGAACTATC	121	*95 °C for 3 min, (95 °C for 30 s, 57 °C for 15 s, 72 °C for 30 s)- 40 cycles.	[18]
*R. albus*	CCCTAAAAGCAGTCTTAGTTCG/ CCTCCTTGCGGTTAGAACA	175	*95 °C for 9 min, (95 °C for 30 s, 55 °C for 30 s, 72 °C for 30 s)- 48 cycles, 72 °C for 10 min.	[20]
*S. bovis*	CTAATACCGCATAACAGCAT/ AGAAACTTCCTATCTCTAGG	869	95 °C for 3 min, (95 °C for 30 s, 57 °C for 30 s, 72 °C for 1 min)- 35 cycles.	[21]
*R. flavefaciens*	CGAACGGAGATAATTTGAGTTTACTTAGG/ CGGTCTCTGTATGTTATGAGGTATTACC	132	*95 °C for 3 min, (95 °C for 15 s, 57 °C for 30 s, 72 °C for 30 s)- 40 cycles.	[18]

**Table 2 microorganisms-08-01605-t002:** The Essential Oil Constituents of Ganghwa (GA), Injin (IN), and San (SA) Korean Native Wormwoods (g kg^−1^, DM).

Item	GA	IN	SA
(-)-Caryophyllene oxide	26.4	38.0	44.3
endo-Borneol	185	ND ^1^	48.2
1,8-Cineole	ND	83.6	56.7
3-Cyclohexen-1-ol	102	ND	ND
Naphthalene (CAS)	44.0	ND	ND
Phenol, 2-methoxy-4-(2-propenyl)- (CAS)	28.6	ND	ND
Bicyclo [2.2.2] oct-2-ene	30.1	ND	ND
Pentan-1,3-Dioldiisobutyrate	511	ND	ND
Caryophyllenol-II	34.2	ND	ND
2-Pentadecanone, 6,10,14-trimethyl- (CAS)	38.7	ND	ND
Benzene, 1-methoxy-4-nitro-(CAS)	ND	ND	25.6
*trans*-Caryophyllene	ND	ND	162
α-Humulene	ND	ND	55.8
Naphthalene	ND	ND	57.7
α-amorphene	ND	ND	31.3
α-Muurolene-(-)	ND	ND	31.9
ɣ-Cadinene	ND	ND	39.7
δ-Cadinene	ND	ND	109
α-Cadinol	ND	ND	131
T-Muurolol	ND	ND	151
Hexadecanoic acid, ethyl ester(CAS)	ND	ND	33.3
Ethyl Linoleate	ND	ND	22.5
Camphor	ND	332	ND
1-Borneol	ND	299	ND
α-Copaene	ND	20.6	ND
Trans(β)-Caryophyllene	ND	107	ND
Germacrene-d	ND	47.6	ND
β-bisabolene	ND	47.6	ND
selin-11-en-4-α-ol	ND	24.6	ND

^1^ ND, not detected.

**Table 3 microorganisms-08-01605-t003:** Effects of Essential Oils from Different Wormwood Species on In Vitro Digestibility and Fermentation Indices of Soybean Meal and Bermudagrass Hay after 72 h of Incubation in Buffered Rumen Fluid.

Item ^1^	Soybean Meal						Bermudagrass Hay					
	CON ^2^	GA	IN	SA	SEM	*p*-Value	CON	GA	IN	SA	SEM	*p*-Value
IVDMD, g kg^−1^ DM	658	642	625	630	19.4	0.486	308 ^b^	323 ^b^	440 ^a^	411 ^a^	26.8	0.004
IVNDFD, g kg^−1^ DM	1000	1000	1000	1000	0.000	1.000	312 ^a^	217 ^b^	370 ^a^	356 ^a^	22.6	0.001
pH	7.42	7.51	7.48	7.49	0.068	0.458	7.20	7.31	7.29	7.17	0.129	0.588
Ammonia-N, mg N dL^−1^	36.6	35.9	35.3	35.2	1.44	0.664	17.4 ^b^	17.0 ^b^	17.9 ^b^	20.5 ^a^	0.26	0.001
Total SCFA, m*M* L^−1^	154.2 ^a^	142.6 ^b^	148.8 ^a,b^	145.7 ^a,b^	3.18	0.026	136.1 ^a^	119.8 ^b^	130.2 ^a^	131.7 ^a^	2.87	0.001
Acetate, mol 100 mol^−1^	58.2	58.0	58.3	57.9	0.85	0.922	67.6	67.7	68.4	68.1	0.72	0.501
Propionate, mol 100 mol^−1^	16.6	17.2	16.3	16.3	0.44	0.097	14.8	15.1	15.4	15.2	0.26	0.157
Isobutyrate, mol 100 mol^−1^	2.95	2.90	2.98	2.94	0.206	0.976	1.84	1.74	1.57	1.57	0.182	0.269
Butyrate, mol 100 mol^−1^	12.9	12.8	13.0	13.0	0.23	0.759	10.2	9.76	9.60	9.87	0.273	0.223
Isovalerate, mol 100 mol^−1^	5.30	5.43	5.37	5.77	0.194	0.072	3.23	3.10	2.70	2.83	0.216	0.579
Valerate, mol 100 mol^−1^	4.17	4.60	4.13	4.20	0.183	0.096	2.23	2.53	2.10	2.37	0.206	0.141
A:P ratio	3.51	3.37	3.58	3.56	0.109	0.156	4.55	4.49	4.45	4.47	0.101	0.649

^a,b^ Means in the same row with different superscripts differ significantly (*p* < 0.05). ^1^ IVDMD, in vitro dry matter digestibility; IVNDFD, in vitro neutral detergent fiber digestibility; SCFA; short-chain fatty acid; A:P ratio, acetate to propionate ratio. ^2^ CON, without essential oil application; GA, Ganghwa essential oil applied; IN, Injin essential oil applied; SA, San essential oil applied.

**Table 4 microorganisms-08-01605-t004:** Effects of San Wormwood Essential Oil Inclusion at Different Levels on In Vitro Digestibility and Fermentation Indices of Soybean Meal and Bermudagrass Hay after 72 h of Incubation in Buffered Rumen Fluid.

Item ^1^	Soybean Meal						Bermudagrass Hay					
	0 ^2^	0.1	1	10	SEM	Effects ^3^	0	0.1	1	10	SEM	Effects
IVDMD, g kg^−1^ DM	743	764	721	725	17.1	†, Q	422 ^b^	391 ^b^	465 ^a,b^	532 ^a^	28.2	**, L
IVNDFD, g kg^−1^ DM	1000	1000	1000	1000	0.000	NS	265 ^b,c^	225 ^c^	317 ^b^	428 ^a^	19.7	*, Q
pH	6.98	7.01	6.90	6.94	0.100	NS	6.77	6.76	6.75	6.73	0.034	NS
Ammonia-N, mg N dL^−1^	98.6 ^b^	111.0 ^a^	101.9 ^a,b^	106.6 ^a,b^	3.05	**, C	50.0	49.9	48.8	49.0	2.06	NS
Total SCFA, m*M* L^−1^	126.3 ^b^	127.0 ^b^	131.3 ^a,b^	135.5 ^a^	2.00	**, L	101.7 ^b^	102.0 ^b^	114.3 ^a^	107.7 ^a,b^	2.97	**, Q
Acetate, mol 100 mol^−1^	53.8	53.5	54.8	55.5	1.06	NS	63.1 ^a^	63.6 ^a^	58.6 ^b^	62.1 ^a^	1.15	**, Q
Propionate, mol 100 mol^−1^	18.9 ^a^	18.3 ^a,b^	17.8 ^a,b^	17.3 ^b^	0.39	*, L	17.9	17.4	18.4	16.8	0.43	†, Q
Isobutyrate, mol 100 mol^−1^	3.71	3.33	3.30	2.66	0.551	NS	2.55 ^b^	2.34 ^b^	3.96 ^a^	2.06 ^b^	0.443	**, Q
Butyrate, mol 100 mol^−1^	11.8	12.1	12.6	12.7	0.44	NS	9.6 ^a,b^	9.4 ^b^	10.4 ^a,b^	11.0 ^a^	0.53	**, L
Isovalerate, mol 100 mol^−1^	6.50 ^a,b^	6.49 ^a,b^	6.39 ^b^	6.88 ^a^	0.165	**, Q	3.94 ^c^	3.94 ^c^	4.71 ^a^	4.43 ^b^	0.050	**, Q
Valerate, mol 100 mol^−1^	5.35	5.73	5.10	4.96	0.581	NS	2.82	3.39	2.89	2.93	0.295	*, C
A:P ratio	2.85 ^b^	3.01 ^a,b^	3.08 ^a,b^	3.22 ^a^	0.071	*, L	3.52 ^a,b^	3.66 ^a^	3.18 ^b^	3.65 ^a^	0.116	**, Q

^a–c^ Means in the same row with different superscripts differ significantly (*p* < 0.05). ^1^ IVDMD, in vitro dry matter digestibility; IVNDFD, in vitro neutral detergent fiber digestibility; SCFA, short-chain fatty acid; A:P ratio, acetate to propionate ratio. ^2^ Application of essential oil at 0, 0.1, 1, and 1 mg/kg per 40 mL of buffered rumen fluid. ^3^ NS = Not significant; L = Linear effect; Q = Quadratic effect; C = Cubic effect; * *p* < 0.05; ** *p* < 0.01; † 0.05 < *p* < 0.10.

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
