# Peer review of "Effects of Wormwood (Artemisia montana) Essential Oils on Digestibility, Fermentation Indices, and Microbial Diversity in the Rumen"

_microorganisms, 2020, doi:10.3390/microorganisms8101605_

Round 1

Reviewer 1 Report

The reviewed manuscript entitled “Effects of wormwood (Artemisia Montana) essential oils on digestibility, fermentation indices, and microbial diversity in the rumen” reports two in vitro experiments: first based on three species of wormwood EO – Ganghwa (GA), Injin (IN), or San (SA); and second, when SA was applied at different levels - 0, 0.1, 1, and 10 ppm. As a incubated substrates bermudagrass or soybean meal were used. Based on the used treatments Authors investigated the effect of wormwood EO or different levels of SA on in vitro ruminal fermentation, digestibility and fibrolytic bacterial counts following bermudagrass or soybean meal fermentation.

The general idea of the reviewed MS is with line of the Microorganisms journal scope and could be potentially interesting for the readers.

Although the presented topic is interesting and the MS shows some scientific potential, I have some methodology doubts that I listed below:

If I properly understand two experiments based on 72 h of incubation were investigated. I kindly ask the Authors to explain why so long incubation period was used. Based on obtained results: mainly pH and total VFA, I suggest to check carefully obtained result, calibration or calculation. The results of pH and VFA in my opinion are outside the range of physiological norm and in vitro conditions. This can deeply influence on microorganisms populations.

Please clearly state how many repetitions of experiment did you prepare? In L 95 is stated: “Each treatment was prepared in triplicate…” In my opinion one repetition of experiment is not enough to create proper numbers of repetition in in vitro experiment– at least it have to be three runs within three consecutive days, each run have to have at least three repetitions of each treatments. Finally not n=3 but at least n=9 have to be.

Other doubts:

In introduction part please introduce new references. To improve novelty it is necessary to use references not only from 2006-2014. It should be mentioned that nowadays, the potential of using traditional medicinal plants including Artemisia Montana in the form of whole plants or some extract to modify rumen fermentation characteristics as a substitute for antimicrobials is of growing interest. It can be helpful to use the following publications: Petrič, et al., 2020. Effect of dry medicinal plants (wormwood, chamomile, fumitory and mallow) on in vitro ruminal antioxidant capacity and fermentation patterns of sheep. J. Anim. Physiol. Anim. Nutr. (Berl). 1219–1232. https://doi.org/10.1111/jpn.13349 or Szulc, et al., 2020. Ruminal fermentation, microbial population and lipid metabolism in gastrointestinal nematode-infected lambs fed a diet supplemented with herbal mixtures. PLoS One 15, 1–26. https://doi.org/10.1371/journal.pone.0231516

Publications by Szulc and Petrič can be also used to improve discussion part, because only one publication is from 2020, none from 2019, 2018. It have to be underlined that researchers are still looking for the most effective sources and doses of plant bioactive substance that could be recommended for improvement of rumen metabolism and rumen microbial populations and finally improve feed efficiency, product quality and decrease negative impact of animal production on environment. In my opinion above things should be included in Introduction part and discussion.

L 64-65 It is state: “The high and low fiber substrates consisted of soybean meal (SBM) and bermudagrass hay (BGH, Cynodon dactylon cultivar Tifton 85), respectively.” – in my opinion there is opposite low and high fiber substrates consisted of soybean meal (SBM) and bermudagrass hay…Please improve it.

Please clearly state why such EO (exp. 1) and levels (0, 0.1, 1, and 10 ppm) of SA were tested.

In conclusion as well as in abstract please clearly state that it is in vitro study. As the Authors know in vitro study is only the first step for verification of stated scientific hypothesis.

Considering my concerns listed above, I do recommend the evaluated MS for publication in Microorganisms but only after major revision and after clarifying all methodology doubts.

Author Response

RESPONSE LETTER

All authors greatly appreciate with the reviewer’s comments, and found them very useful for improving the quality of this manuscript. The responses to the comments are as follows:

  1. If I properly understand two experiments based on 72 h of incubation were investigated. I kindly ask the Authors to explain why so long incubation period was used. Based on obtained results: mainly pH and total VFA, I suggest to check carefully obtained result, calibration or calculation. The results of pH and VFA in my opinion are outside the range of physiological norm and in vitro conditions. This can deeply influence on microorganisms populations.

Response: Thanks for your constructive comments. Yes, we agree that 72 h of incubation seems so long for SBM substrate, but not for BGH substrate. Generally, roughage sources were incubated with rumen buffer for 72-96 h in many studies to provide sufficient time for the fermentation and digestion of fibrous constituents (Zhao et al., 2020 & Chen et al., 2008 published at Animal Feed Science and Technology; Iqbal et al., 2018 published at Journal of Animal Nutrition). The 72 h incubation in this study was used for both substrates to ensure the same incubation conditions were used and to allow the results to be compared. Regrading pH and total VFA concentration, these results in this study are similar to the previous studies (Weinberg et al., 2003 & Weinberg et al., 2004 published at Journal of Dairy Science; Benchaar et al., 2007 published at Canadian Journal of Animal Science; Hartinger et al., 2019 published at Animal Feed Science and Technology). Therefore, the pH and total VFA concentration in this study are in normal range for in vitro study.

  1. Please clearly state how many repetitions of experiment did you prepare? In L 95 is stated: “Each treatment was prepared in triplicate…” In my opinion one repetition of experiment is not enough to create proper numbers of repetition in in vitro experiment– at least it have to be three runs within three consecutive days, each run have to have at least three repetitions of each treatments. Finally not n=3 but at least n=9 have to be.

Response: Yes, we agree with your comment, regarding repeating the experiment on 3 consecutive days. This would have been ideal unfortunately it was not done in this case. Although some studies have been published without repetition of in vitro assays across days (Weinberg et al., 2004 published at Journal of Dairy Science; Yuan et al. 2015 published at Animal Feed Science and Technology), we agree that the best practice is to repeat the run on 3 consecutive days and now routinely do this. Unfortunately, this study was conducted several years ago before we adopted the ideal practice.

  1. In introduction part please introduce new references. To improve novelty it is necessary to use references not only from 2006-2014. It should be mentioned that nowadays, the potential of using traditional medicinal plants including Artemisia Montana in the form of whole plants or some extract to modify rumen fermentation characteristics as a substitute for antimicrobials is of growing interest. It can be helpful to use the following publications: Petrič, et al., 2020. Effect of dry medicinal plants (wormwood, chamomile, fumitory and mallow) on in vitro ruminal antioxidant capacity and fermentation patterns of sheep. J. Anim. Physiol. Anim. Nutr. (Berl). 1219–1232. https://doi.org/10.1111/jpn.13349 or Szulc, et al., 2020. Ruminal fermentation, microbial population and lipid metabolism in gastrointestinal nematode-infected lambs fed a diet supplemented with herbal mixtures. PLoS One 15, 1–26. https://doi.org/10.1371/journal.pone.0231516. Publications by Szulc and Petrič can be also used to improve discussion part, because only one publication is from 2020, none from 2019, 2018. It have to be underlined that researchers are still looking for the most effective sources and doses of plant bioactive substance that could be recommended for improvement of rumen metabolism and rumen microbial populations and finally improve feed efficiency, product quality and decrease negative impact of animal production on environment. In my opinion above things should be included in Introduction part and discussion.

Response: These are valid and important points. We revised the introduction as you recommended.

  1. L 64-65 It is state: “The high and low fiber substrates consisted of soybean meal (SBM) and bermudagrass hay (BGH, Cynodon dactylon cultivar Tifton 85), respectively.” – in my opinion there is opposite low and high fiber substrates consisted of soybean meal (SBM) and bermudagrass hay…Please improve it.

Response: Corrected.

  1. Please clearly state why such EO (exp. 1) and levels (0, 0.1, 1, and 10 ppm) of SA were tested.

Response: Clarified at introduction (L 55-62)

  1. In conclusion as well as in abstract please clearly state that it is in vitro study. As the Authors know in vitro study is only the first step for verification of stated scientific hypothesis.

Response: Yes, we agree. Now stated clearly.

Reviewer 2 Report

This research is about the effect of essential oil from wormwood on the rumen microbial populations. There are several major revisions and this manuscript should not be accepted by Microorganisms by the Reviewer's opinion.

  1. First of all, this research was only carried out the in vitro test. Although experiments on ruminants are not easy to perform, the results of in vitro experiments do not represent the true reactions in animals. The reviewer suggests that simple animal tests can be carried out to confirm the test results.

  1. There are too many grammar errors to understand what the authors want to describe. Please modify the manuscript by native English speakers for a better reading experience.

  1. The author should display the figures of different scales separately to prevent the effect of each treatment group from becoming unobvious.

Author Response

RESPONSE LETTER

All authors greatly appreciate with the reviewer’s comments, and found them very useful for improving the quality of this manuscript. The responses to the comments are as follows:

  1. First of all, this research was only carried out the in vitro test. Although experiments on ruminants are not easy to perform, the results of in vitro experiments do not represent the true reactions in animals. The reviewer suggests that simple animal tests can be carried out to confirm the test results.

Response: We agree with your comment that in vivo experiments are more accurate to determine treatment responses. This experiment consisted of 16 treatments from experiment 1 and 2. Conducting one or two animal trials with so many treatments will be very expensive, protracted and labor intensive. This is why the in vitro approach was used. Future animal studies will be conducted to validate the most promising treatments from the in vitro study.

  1. There are too many grammar errors to understand what the authors want to describe. Please modify the manuscript by native English speakers for a better reading experience.

Response: The paper has been thoroughly revised by a native English speaker.

  1. The author should display the figures of different scales separately to prevent the effect of each treatment group from becoming unobvious

Response: Corrected by creating several different graphs.

Reviewer 3 Report

File is attached!

Author Response

RESPONSE LETTER

All authors greatly appreciate with the reviewer’s comments, and found them very useful for improving the quality of this manuscript. The responses to the comments are as follows:

  1. Line 16: instead of Soybean, soybean (not capital letter). Instead of measurement unit ppm, mg/kg (μg/g) is much better. The change of ppm to this form is required all over the manuscript.

Response: Corrected.

  1. Lines 20 and 235: Never start sentences with abbreviation or numbers.

Response: Corrected.

  1. Line 21 and all over the manuscript: The use of total short chain fatty acids (TSCFA) is better than TVFA. Chang VFA to SCFA everywhere in the manuscript.

Response: Corrected.

  1. Line 22: …numbers of Ruminococcus albus and Streptococcus…

Response: Corrected.

  1. Line 27: …rumen fermentation…

Response: Corrected.

  1. Line 56: in vitro study…

Response: Corrected.

  1. Line 82: Describe the age of the cows!

Response: Described it.

  1. Lines 91, 123 and 179: …in Experiment 1 or 2… (with capitals).

Response: Corrected.

  1. Line 99: …rumen buffer was filtered…

Response: Corrected.

  1. Lines 147-146: In the text of chapter “Results” no numerical description of results in brackets is required. Describe only the tendencies of changes in the text. The numerical description of results is contained in Table 3.

Response: Deleted

  1. Line 182: …acetate : propionate (A : P ) ratio…

Response: Corrected.

  1. Line 197: …SA… application at any different levels.

Response: Corrected.

  1. Line 245: The α-cadinol…

Response: Corrected.

Round 2

Reviewer 1 Report

Dear Authors

Thank you for mostly improving all my previous doubts. I have another small request to mention in discussion part that others Authors also obtained high concentration of SCFA and pH value. Just to clarified your results in case of in vitro studies. Thank you. 

Author Response

Thank you for mostly improving all my previous doubts. I have another small request to mention in discussion part that others Authors also obtained high concentration of SCFA and pH value. Just to clarified your results in case of in vitro studies. Thank you. 

Response: Clarified at L222-223.

Reviewer 2 Report

Its OK for publish.

Author Response

Its OK for publish.

Response: Thanks for your valuable comments and helps for improving this manuscript.